# Caregiving + Migrant Background = Double Jeopardy? Associations between Caregiving and Physical and Psychological Health According to Migrant Backgrounds in Norway

**DOI:** 10.3390/ijerph20105800

**Published:** 2023-05-12

**Authors:** Kamila Hynek, Aslaug Gotehus, Fredrik Methi, Ragnhild Bang Nes, Vegard Skirbekk, Thomas Hansen

**Affiliations:** 1Department of Mental Health and Suicide, Norwegian Institute of Public Health, 0213 Oslo, Norway; kamila.a.hynek@gmail.com (K.H.); ragnhildbang.nes@fhi.no (R.B.N.); 2Work Research Institute (AFI), Oslo Metropolitan University, 0130 Oslo, Norway; aslag@oslomet.no; 3Department of Health Services Research, Norwegian Institute of Public Health, 0213 Oslo, Norway; fredrik.methi@fhi.no; 4Promenta Research Center, University of Oslo, 0373 Oslo, Norway; 5Department of Philosophy, Classics, History of Arts and Ideas, University of Oslo, 0213 Oslo, Norway; 6Center for Fertility and Health, Norwegian Institute of Public Health, 0213 Oslo, Norway; vegard.skirbekk@fhi.no; 7Department of Psychology, University of Oslo, 0373 Oslo, Norway; 8Norwegian Social Research (NOVA), Oslo Metropolitan University, 0130 Oslo, Norway

**Keywords:** informal care, health, migrant background, well-being, mental distress, Norway

## Abstract

Informal caregiving can have detrimental consequences for physical and psychological health, but the impacts are highly heterogenous. A largely ignored question is whether these impacts differ with migrant backgrounds, and whether caregiving and a migrant background combine to create double jeopardy. We explored these questions using large-scale data that allows stratification by sex, regional background, and types (inside vs. outside of household) of caregivers. We used cross-sectional 2021 data collected from two Norwegian counties as part of the Norwegian Counties Public Health Survey (N = 133,705, RR = 43%, age 18+). The outcomes include subjective health, mental health, and subjective well-being. The findings show that both caregiving, especially in-household caregiving, and a migrant background relate to lower physical–psychological health. In bivariate analysis, non-Western caregivers, women particularly, reported poorer mental health and subjective well-being (but not physical health) than other caregiver groups. After controlling for background characteristics, however, no interaction exists between caregiver status and migrant background status. Although the evidence does not suggest double jeopardy for migrant caregivers, caution is warranted due to the likely underrepresentation of the most vulnerable caregivers of migrant backgrounds. Continued surveillance of caregiver burden and distress among people of migrant backgrounds is critical to develop successful preventive and supportive intervention strategies for this group, yet this aim hinges on a more inclusive representation of minorities in future surveys.

## 1. Introduction

Against a background of population aging, many governments aim to promote and support informal care, i.e., unpaid help provided by relatives or others to people who need help because of health problems [1]. Much research demonstrates that such care can be physically, mentally, and financially challenging, and it has been linked with myriad adverse physical and psychological health outcomes [2,3,4,5]. That said, these impacts are highly heterogeneous, depending on personal and care-related circumstances as well as cultural and institutional frameworks. For example, these impacts are especially pronounced among long-term and high-intensity personal carers [6,7,8]. Additionally, in-household (in-hh) caregiving is associated with more negative consequences than out-of-household (out-hh) caregiving [9,10]. Further, women seem to experience stronger health deterioration than men as a result of such caregiving [11,12,13]. 

A largely overlooked issue is whether caregiving has distinct health consequences for people with migrant backgrounds. Many countries are facing an increasing number of individuals with personal or parental histories of migration, hereafter individuals with migrant backgrounds, and consequently, these countries have increasing numbers of caregivers with migrant backgrounds. This calls for an investigation of the potential differences in the impact of informal care on health outcomes among individuals with and without migrant backgrounds.

As caregivers, individuals who are migrants or of migrant backgrounds may pose unique risks. One reason may be that individuals of particularly non-Western migrant backgrounds typically report higher social expectations of family care (filial responsibility norms) and less use of formal care services (e.g., home care, respite care, and nursing homes) [14,15]. In the Nordic countries, it has been shown that relatively few older members from ethnic minority groups live in long-term care facilities such as nursing homes [16], and they are likely to rely on their family for informal care [17,18]. Lower utilization of formal care services reflects factors such as the inability to perceive and acknowledge a need for health care, low health literacy, a lack of knowledge about options in health care, and a preference for informal care as well as feelings of familial, cultural, or religious responsibility to provide care (ibid.). Obligation towards care provision to the elderly, especially for women, has been suggested as another reason [14,15,19,20]. Further, this lower utilization may also stem from a perceived lack of cultural sensitivity and adaptation in nursing homes for minority patients such as linguistic barriers and limited use of interpreters, different food habits, and limited opportunities to perform religious practices [14,21,22]. However, it is important to stress that the utilization of and attitudes towards public care services for migrants can be influenced and changed as a result of exposure to the culture of the receiving country [23]. Thus, utilization changes over time, that is, the longer the residency, the higher the use of public care services [24]. 

The relatively sparse existing empirical literature suggests that migrant caregivers report poorer physical–psychological health than their non-migrant counterparts, yet most of these studies originate from the U.S. For example, drawing on the literature of ethnic differences, a meta-analysis of U.S. studies shows that Black American caregivers are less depressed and Asian-American caregivers more depressed than their white American counterparts [25]. The poorer mental health of Asian-American care providers was suggested to reflect a mismatch between the care expectations of the family and individual ambitions and preferences [25]. Similarly, data from the U.S., Canada [26], and Germany [27] suggest that the health consequences of informal caregiving are greater for migrants than for non-migrants. However, much is still unknown about the potentially unique challenges and rewards of caregiving among migrant groups, with little European, and particularly Nordic, evidence.

Caregiving can also be especially demanding for migrants who face other challenges and risks. For example, international evidence shows that migrants are overrepresented in terms of poor physical and mental health, poor living and working conditions, linguistic barriers, and difficulties adapting to the new society and culture, and many experience obstacles such as racism and discrimination [28,29,30,31]. These patterns are echoed in Norwegian official statistics and research, showing that the migrant population, compared with non-migrants, displays or reports disadvantages in terms of educational attainment [32], workforce participation [33], income [34], and physical and mental health [35], and also that they use healthcare services to a lower extent [24]. Thus, migrants have a higher risk profile in terms of the known and likely predictors of high caregiver distress (e.g., [25,26,36,37,38,39]). 

Further, although caregiving continues to be a gendered activity, with women being more likely to provide care, there may be differences between non-migrants and migrants and across subgroups of migrants, e.g., non-Western and Western migrants. Overall, studies suggest that migrant women from non-Western countries are confronted with stronger familial and cultural expectations to provide care (e.g., [15,40,41,42]). At the same time, these migrant women may face expectations regarding integration and social participation in the country of residency. This potentially results in many being in an “in between” position, facing contradictory expectations from the family on one side and their new society on the other. 

A timely and important question is, therefore, whether individuals with migrant backgrounds, i.e., with personal or parental histories of migration, are victims of “double jeopardy”: disadvantaged by being informal caregivers and by being migrants [30]. Clarifying the health-related costs of informal caregiving affecting the growing segment of caregivers with a migrant background is, thus, a critical step toward enhanced and targeted support for a potentially vulnerable but growing group. This study aims to enhance our understanding by mapping and comparing characteristics and health-related associations of informal caregiving across migrants and non-migrants in Norway. We contextualized all analysis by sex, regional (Western vs. non-Western) migrant background, and type of caregiving (in-household vs. out-of-household). 

## 2. Materials and Methods

This study used combined data from two cross-sectional surveys of The Norwegian Counties Public Health Survey (NCPHS), which was conducted in Møre and Romsdal counties during 1–14 February 2021 [43] and in Viken county during 8–26 November 2021 [44]. A random sample of residents aged 18+ extracted from the National Population Register (NPR) were invited to participate through an email and/or SMS message with a link to the online survey. Both phone numbers and email addresses were extracted from the Norwegian Digital Agency Register. All participants signed an informed consent form to participate. The number of invited individuals in Møre and Romsdal counties was 54,465, out of whom 45.8% responded (n = 24,967). In Viken county, the number of invited individuals was 265,684, of whom 40.9% responded (n = 108,738). The total number of individuals included in the present study was 133,705. After listwise deletion, the analytical sample comprised 129,862 individuals. 

### 2.1. Variables

#### 2.1.1. Dependent Variables

Mental health problems were assessed using the five-item Hopkins Symptom Checklist (HSCL-5). We calculated a mean score for respondents who answered at least three out of five questions. Self-perceived health was measured by asking a single question, “How do you consider your health at the moment”, with the response options ranging from 0 to 3: “poor”, “neither good nor poor”, “good”, and “very good”. Psychological well-being was measured by the mean of three items (engaged, happy, satisfied with your life) all rated on a scale from 0 to 10 (α = 0.74).

#### 2.1.2. Caregiving Variables

Informal caregiving was defined as providing regular unpaid care to a person who is ill, disabled, or old. Response alternatives were: “no” (0), “yes, to one person” (1), and “yes, to two or more individuals” (2). This variable was dichotomized into “yes” (1 or 2) and “no”. Caregivers were asked whether the care they provide is to someone inside or someone outside of the household. Based on the response, caregivers were divided into in-household (in-hh) and out-of-household (out-hh) care providers. As in-hh generally is more demanding (see Section 1), those providing care both in-hh and out-hh were coded as in-hh. 

#### 2.1.3. Migrant Background

Individuals responding “yes” to “Are you or any of your parents born abroad” were considered to be “with migrant background”, whereas those responding “no” were categorized as being without migrant background, hereafter termed “non-migrants”. Based on a follow-up question to the former group about their region of origin, we divided migrants into “Western” (predefined categories: Nordic, Western Europe, Eastern Europe, and North America and Oceania) and “non-Western” (Africa, West Asia, East Asia, and Latin America) migrants. As we are unlikely to have respondents from other Oceanic countries than Australia and New Zealand, our categorization of Western and non-Western (all except Western Europe, North America, Australia, and New Zealand) aligns with that used previously by national statistical agencies [45]. 

#### 2.1.4. Control Variables

Information on sex and age was extracted from NPR. Sex is a dichotomous variable divided into male and female. Age was categorized into age groups 18–29, 30–39, 40–49, 50–59, 60–69, and 70+. This crude categorization (age and sex/gender) was applied by the data providers for anonymization purposes. Education was divided into “upper-secondary education or lower” (hereafter ≤ Upper-secondary) and “at least university education” (hereafter Higher). Marital status was coded into three categories: single, married/cohabiting, and having a non-cohabiting partner. Employment status was divided into employed (full- and part-time employed, self-employed, or sick leave), non-employed (unemployed, disability pension, or social welfare benefit) and other (retired, home worker, student, or military service). 

### 2.2. Analytical Strategy

Descriptive statistics were performed to report the characteristics of the study sample by sex and migrant background. We used linear regression to study differences between the groups. We adjusted for age, marital status, education, and employment status, and we stratified all analyses by sex. In the first model, we regressed caregiver status and migrant background on outcome variables. In Model 2, we added interaction terms between caregiving and migrant background, and we used these analyses to plot point estimates into Figure 1, Figure 2 and Figure 3. All analyses were stratified by sex and conducted in Stata SE. 17 (StataCorp, College Station, TX, USA) and Rstudio with R version 4.2.2.

## 3. Results

### Descriptive Statistics

Table 1 presents characteristics of the study population by sex and migrant background. Caregiving to someone in the household was about twice as common among male (6.0%) and female (5.7%) non-Western migrants as among their respective non-migrant (2.7% and 3.1%) and Western migrant background (3.3% and 3.8%) counterparts. Out-hh caregiving, however, was more common among non-migrant men and women. The table also shows that individuals of especially non-Western migrant backgrounds were disproportionately younger, and thus, they were slightly more likely to have higher education and be employed than non-migrants. 

Figure 1, Figure 2 and Figure 3 present mean levels of physical health, mental health, and psychological well-being by caregiver status, migrant background, and sex. In general, we found that in-hh caregivers reported the most adverse levels, and levels among non-caregivers and out-hh caregivers tended to be quite similar and not significantly different. Between the two sexes, there are similarities and differences. For self-perceived health, we found few gender differences, except that women reported better health than men among two groups of non-migrants: non-caregivers and out-hh caregivers (*p*’s < 0.05). For mental health, women generally reported more problems than men did (*p* < 0.05), except in four subgroups (Western and non-Western out-hh caregivers and in-hh caregivers). An opposite pattern emerged for psychological well-being, with women reporting significantly (*p* < 0.05) higher levels than men in all groups except among Western and non-Western out-hh caregivers and in-hh caregivers. However, the low numbers, especially among migrant caregivers, make the estimates uncertain. 

Regarding the interaction between caregiver and migrant background statuses for men and women, patterns differ for self-reported health vis-à-vis mental health and psychological well-being. Regarding the former (Figure 1), and among both sexes, we found that non-caregivers reported slightly but not significantly (*p* > 0.05) better health than out-hh caregivers and better health than in-hh caregivers, but the latter difference is only significant (*p* < 0.05) for non-migrants and Western migrants. 

Regarding mental health (Figure 2), non-caregivers reported slightly better mental health than out-hh caregivers, but the difference is only significant (*p* < 0.05) among non-migrants (both sexes), male Western migrants, and female non-Western migrants. Non-caregivers reported better mental health than in-hh caregivers across all sexes and migrant status groups (*p* < 0.05 for both). Out-hh caregivers reported better mental health than in-hh caregivers, and the differences are significant, except among male Western migrants and non-Western migrants (both sexes) (*p* > 0.05). 

Regarding psychological well-being (Figure 3) among non-migrants and both sexes, out-hh caregivers reported higher well-being than non-caregivers, which in turn reported higher well-being than in-hh caregivers, and all differences are significant (*p* < 0.05). Among Western migrants, the differences between non-caregivers and out-hh caregivers are not significant, but both groups reported higher well-being than in-hh caregivers (*p* < 0.05 only among women). Among male and female non-Western migrants, non-caregivers and out-hh caregivers reported non-significantly different levels of well-being, but both groups reported higher well-being than in-hh caregivers. 

The patterns in Figure 1, Figure 2 and Figure 3 are also quite consistent across counties and age groups (18–39, 40–59, and 60+) (ancillary analyses, not shown).

The results from the linear regression analyses are presented in Table 2. Model 1 shows that both in-hh and out-hh caregiving are associated with poorer subjective health and mental health, regardless of sex. An exception is that out-hh caregiving is unrelated to self-reported health among men. Furthermore, although in-hh caregiving is associated with lower psychological well-being, out-hh caregiving is associated with higher psychological well-being for both sexes. 

Moreover, although self-reported health was rather similar across migrant status groups, there was a gradually decreasing gradient in mental health and psychological well-being from non-migrants to Western migrants to non-Western migrants, although most differences are not statistically significant. More specifically, self-reported health was slightly lower among female Western migrants than female non-migrants, whereas other migrant status group differences in health are not significant (*p* > 0.05). With respect to psychological well-being, and for both genders, non-migrants reported higher levels than Western migrants, who in turn reported higher levels than non-Western migrants (all with *p* < 0.05, except for the difference between female non-migrants and female Western migrants, which has *p* > 0.05). Turning to mental health among women, non-migrants reported better mental health than both migrant groups (*p* < 0.05), and Western migrants reported slightly but not significantly better mental health that non-Western migrants. Among men, non-migrants reported better mental health than Western migrants, who in turn reported better mental health than non-Western migrants (all with *p* < 0.05). 

In Model 2, interaction terms between caregiver status and migrant background were added. With both sexes, we found no significant differences in the association between self-perceived health and informal caregiving by migrant background. Further, we found that out-of-household caregiving is associated with poorer mental health for women with non-Western migrant backgrounds but with poorer psychological well-being for women with Western migrant backgrounds. 

## 4. Discussion

In this study, we explored and compared associations between informal caregiving and health outcomes among men and women of non-migrant and migrant background. We found that informal caregiving is slightly more common among individuals of non-migrant than of migrant backgrounds. The latter group’s lower likelihood of having family and especially older parents in Norway may, at least partly, explain this pattern. However, caregiving to someone in the household is about twice as common among both male and female non-Western migrants than among those of non-migrant and Western-migrant backgrounds. Cultural differences in filial responsibility norms and attitudes to formal care could explain this difference. Surprisingly, among migrants from non-Western countries, a slightly higher percentage of men than women reported providing informal care both inside and outside of the household. This could reflect a higher percentage of migrant men being proficient in Norwegian or English, a skill required to complete the questionnaire. A recent study showed that women, and particularly those who were married, had poorer language proficiency in their second language compared to both unmarried women and married and unmarried men [46]. 

Furthermore, these findings indicate that individuals with non-Western migrant backgrounds in particular reported poorer mental health and psychological well-being than those without a migrant background. Similarly, caregivers—but mainly those who provide care to someone in their household—reported poorer health and psychological well-being than others. 

Is there a double jeopardy effect for caregivers with a migrant background? Although a slight such effect emerges without control for compositional differences, it dissipates in a multivariate context. That is, before control, we found the most adverse levels of mental health and psychological well-being among non-Western in-hh caregivers, yet the differences are slight and not statistically significant. However, when we accounted for the fact that those of migrant backgrounds were markedly younger than others, the associations between caregiving and physical–psychological health were similar regardless of migrant background. These findings contrast with those of previous studies in the literature that support a double jeopardy hypothesis [25,26,27]. Various aspects of the Norwegian welfare system (e.g., support for work–family balance, formal care), which are shown to predict less caregiver distress in Norway than elsewhere [9], could explain the seemingly similar pattern for those of migrant backgrounds in Norway as well. However, this notion remains speculative before we have data on the use and benefits of the relevant formal resources and services to caregivers of migrant and non-migrant backgrounds. Based on qualitative studies, we know that migrants generally use formal care services and health services to a lesser extent than non-migrants in Norway [14,15]. Thus, one may assume and anticipate that migrants benefit less from the national welfare system. 

There are several potential caveats and explanations to the notion that caregiving carries similar health implications irrespective of migrant background. First, the data used do not provide information about whether each individual has family in Norway and potentially someone to care for. The lack of family in Norway could depress the rates of caregiving for those of migrant backgrounds. In addition, we lack information about transnational care. Many migrants may provide financial or other types of care to family members in their home country, solely or in combination with care in Norway, and they may experience challenges and strains as a result [47]. Second, data was collected during the pandemic, an unusual time when potential differences in health outcomes could have decreased between the relevant groups. For example, the burden of caregiving for non-migrant caregivers could have increased due to restricted formal services. Migrants, as discussed, use such services to a lesser extent (e.g., [16,18]). In addition, data were collected during different months of 2021 (February and November) in the two counties. As vaccination rates were highest in late 2021, there could have been a relative return to normality that impacted the results. However, the fact that both months are characterized as relative peak stages of the pandemic in Norway [48], and that the results are similar across the two counties, suggest that the timing of data collection was not impactful. Finally, the surveys were only available in Norwegian; thus, migrants with poor language proficiency were most likely not reached with this survey. It is therefore reasonable to assume that many caregivers with migrant backgrounds were not reached. 

These considerations aside, individuals of migrant backgrounds did report more adverse experiences across all caregiver groups. For example, in-household caregivers with non-migrant backgrounds scored around 6.8 on the 0–10 psychological well-being scale. Hence, if we assume that the effect of in-household caregiving on well-being is mainly causal, then the lower scores in well-being among those of non-Western migrant backgrounds are important because of their already lower well-being. Although the magnitude of the “declines” is similar across the groups, they affect non-migrant caregivers more: “a falling tide sinks all boats”, yet the implications are graver for those lower on the well-being ladder.

This study has several strengths and limitations. A clear strength is the large sample size and high response rate (43.3%). These strongholds enable analysis of subgroups of sufficient size and quality. Further strengths include the range of outcome variables and the recent data. Among the limitations, caution is warranted, as the cross-sectional design precludes conclusions about causality. Further, as the data was collected in two counties only, our sample may not be representative for the whole country. However, Viken county is the most populous county in Norway, and its proportion of individuals with a migrant background is nearly identical to that of the whole country. Furthermore, we have crude information about caregiving, and we lack information about types of care, care recipients, and the duration and intensity of the care provided. Impacts are known to be especially pronounced within long-term and high-intensity personal care [6,7,8]. The information about migrants is similarly crude. The categories of Western and non-Western backgrounds conceal much heterogeneity. However, the relatively small number of individuals with migrant backgrounds who provide care did not allow finer division. Additionally, we are unable to distinguish between generations of migrants, who may face distinct challenges as caregivers. Descendants of migrants may face caregiver distress as they juggle the expectations of Norwegian society and those of older first-generation relatives for whom they provide care and whose expectations of family care often mirror those in their families’ countries of origin [49]. First-generation immigrant caregivers may face their own challenges, as they may be less acculturated or integrated into society than second and higher generations [50,51], and they may have limited knowledge of the Norwegian language and available services [52]. 

## 5. Conclusions

In conclusion, we found similar associations between caregiving and physical–psychological health irrespective of migrant background. It is mainly resident and non-resident caregiving that relates to adverse outcomes. Although the relationships are consistent across groups, they nonetheless seem more serious for non-Western migrant caregivers, who reported lower mental health and psychological well-being overall. That said, the impacts seem milder than suggested by prior research. The contrast could stem from country differences, thereby highlighting the need to replicate our analysis in countries with fewer social protections and less access to formal support. Understanding how migrants react to the difficulties imposed by informal caregiving is essential to support at-risk caregivers and, by extension, their care recipients. This importance is highlighted also by the well-established consequences of compromised psychological health and well-being on daily functioning, prosocial behavior, and physical health [52]. These effects in turn impact caregivers’ ability to provide care, and they increase the risk of institutionalization and of additional health and social costs [53]. Efforts to reduce the burdens of caregiving thus, especially when care coincides with other difficulties, have clear implications for the health and functioning of people in and around the care relationships as well as for wider society. 

## Figures and Tables

**Figure 1 ijerph-20-05800-f001:**
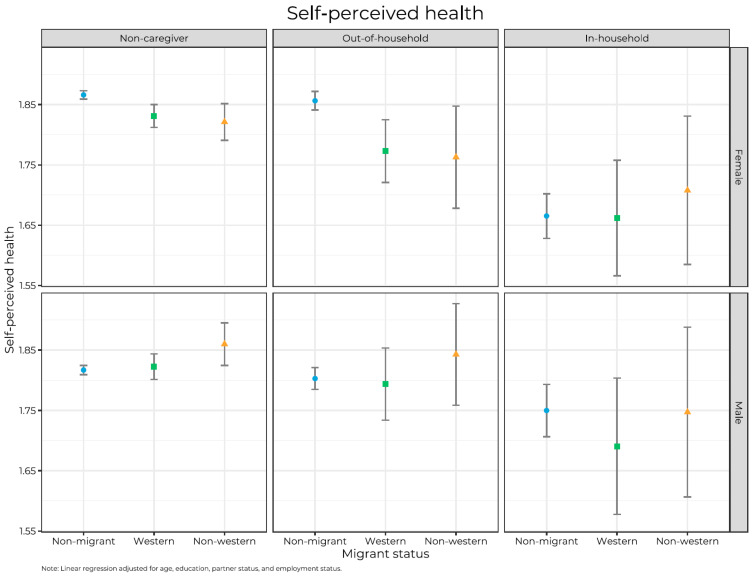
Point estimates of self-perceived health with 95% confidence intervals by caregiver status, migrant background, and sex. Lower values mean poorer self-perceived health.

**Figure 2 ijerph-20-05800-f002:**
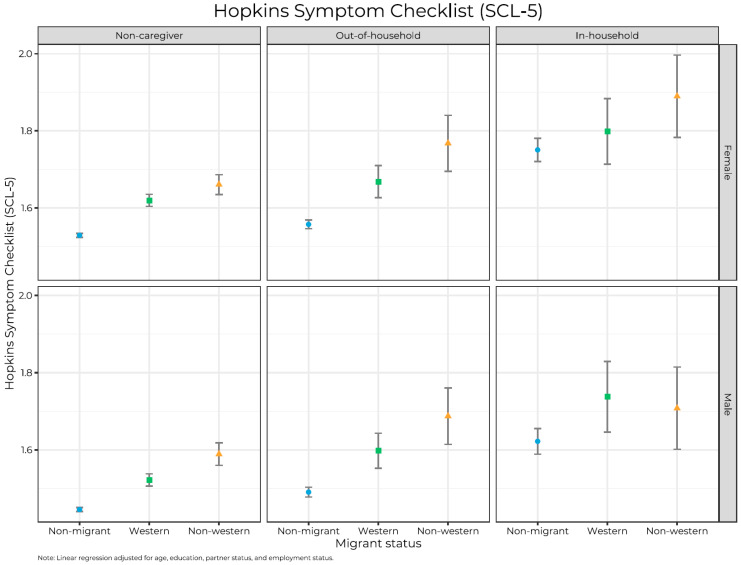
Mean scores of mental health with 95% confidence intervals by caregiver status, migrant background, and sex. Higher values mean poorer mental health.

**Figure 3 ijerph-20-05800-f003:**
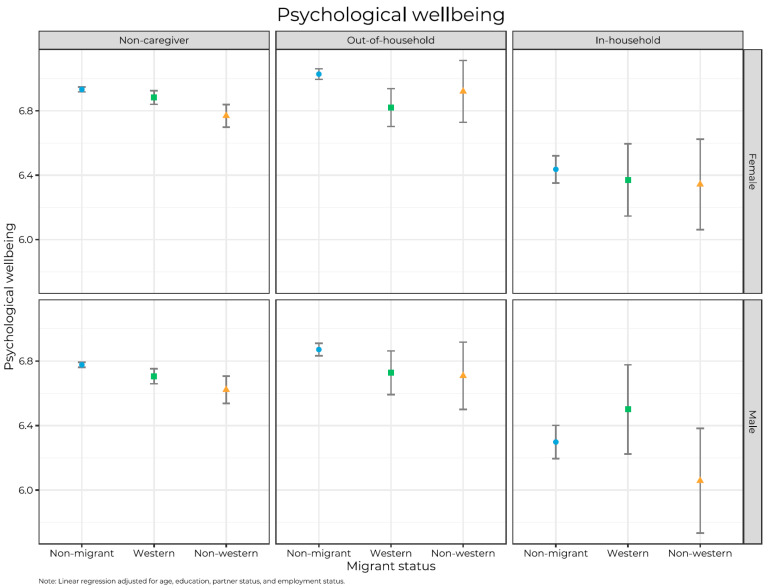
Mean scores of psychological wellbeing with 95% confidence intervals by caregiver status, migrant background, and sex.

**Table 1 ijerph-20-05800-t001:** Characteristics of the study population by sex and migrant background. Data represented as percentages.

	Female, N = 70,771			Male, N = 59,091			Total
	Non-Migrant	Western Migrant	Non-Western Migrant	Non-Migrant	Western Migrant	Non-Western Migrant	
**N**	59,604	7731	3436	50,330	6128	2633	129,862
Non-caregiver	80.4	84.3	83.0	83.4	85.9	81.7	82.1
In-household caregiver	3.1	3.8	5.7	2.7	3.3	6.0	3.2
Out-of-household caregiver	16.5	11.9	11.4	13.9	10.8	12.3	14.7
**Age**							
18–29	11.6	13.4	23.4	8.8	10.3	20.4	11.1
30–39	14.0	21.5	29.5	11.0	18.6	26.6	14.2
40–49	18.4	24.5	26.9	16.6	23.8	25.3	18.7
50–59	23.3	21.1	14.0	23.0	23.2	15.8	22.6
60–69	19.7	12.6	5.3	22.0	15.3	8.8	19.3
70+	13.0	6.9	1.0	18.6	8.8	3.1	14.1
**Education**							
≤Upper-secondary	44.3	32.7	42.2	50.8	44.5	49.6	46.2
Higher	55.7	67.3	57.8	49.2	55.5	50.4	53.8
**Employment status**							
Non-employed	37.6	27.7	22.9	35.2	23.4	22.1	34.7
Employed	56.4	65.0	62.0	61.5	72.5	69.8	60.1
Other	6.0	7.3	15.1	3.3	4.1	8.1	5.2
**Marital status**							
Single	23.0	20.9	24.7	17.8	17.0	25.6	20.7
Married/cohabiting	70.0	72.6	67.2	75.3	76.1	66.9	72.3
Non-cohabiting partner	7.0	6.5	8.1	6.9	6.9	7.5	7.0

**Table 2 ijerph-20-05800-t002:** Linear regression models on the association between caregiver status and three health outcomes: self-perceived health, psychological well-being, and mental health.

	*Self-Perceived Health*				*Psychological Well-Being*				*Mental Health Problems*			
	Female		Male		Female		Male		Female		Male	
	Model 1	Model 2	Model 1	Model 2	Model 1	Model 2	Model 1	Model 2	Model 1	Model 2	Model 1	Model 2
	Coef.	Coef.	Coef.	Coef.	Coef.	Coef.	Coef.	Coef.	Coef.	Coef.	Coef.	Coef.
Caregiver status (ref. non-caregiver)												
In-household	−0.19 ***	−0.20 ***	−0.08 ***	−0.07 **	−0.50 ***	−0.51 ***	−0.45 ***	−0.47 ***	0.22***	0.23 ***	0.17 ***	0.17 ***
Out-of-household	−0.03 **	−0.02 *	−0.01	−0.001	0.07 ***	0.08 ***	0.10 ***	0.11 ***	0.04 ***	0.04 ***	0.04 ***	0.04 ***
Migrant background (ref. non-migrant)												
Western	−0.03 **	−0.03 *	−0.01	−0.01	−0.07 **	−0.04	−0.08 ***	−0.09 ***	0.09 ***	0.09 ***	0.09 ***	0.08 ***
Non-Western	−0.02	−0.02	−0.00	0.00	−0.12 ***	−0.13 ***	−0.20 ***	−0.19 ***	0.12 ***	0.11 ***	0.17 ***	0.17 ***
Interaction care x migrant												
In-hh x Western		0.03		−0.07		−0.02		0.26		−0.04		0.05
In-hh x Non-Western		0.09		−0.05		0.07		−0.09		0.01		−0.06
Out-hh x Western		−0.05		−0.01		−0.16 *		−0.07		0.02		0.03
Out-hh x Non-Western		−0.05		0.001		0.05		−0.01		0.08 *		0.05
Age (ref. 18–29)												
30–39	−0.03 *	−0.03 *	−0.11 ***	−0.11 ***	0.24 ***	0.24 ***	−0.14 ***	−0.14 ***	−0.21 ***	−0.21 ***	−0.06 ***	−0.06 ***
40–49	−0.07 ***	−0.07 ***	−0.20 ***	−0.20 ***	0.36 ***	0.36 ***	−0.14 ***	−0.14 ***	−0.34 ***	−0.34 ***	−0.14 ***	−0.14 ***
50–59	−0.02	−0.02	−0.19 ***	−0.19 ***	0.62 ***	0.62 ***	0.14 ***	0.14 ***	−0.44 ***	−0.44 ***	−0.22 ***	−0.22 ***
60–69	0.31 ***	0.30 ***	−0.02	−0.02	1.21 ***	1.21 ***	0.73 ***	0.73 ***	−0.66 ***	−0.66 ***	−0.42 ***	−0.42 ***
70+	0.58 ***	0.58 ***	0.17 ***	0.17 ***	1.65 ***	1.65 ***	1.15 ***	1.15 ***	−0.86 ***	−0.86 ***	−0.62 ***	−0.62 ***
Education (ref. ≤ upper-secondary)	0.17 ***	0.17 ***	0.19 ***	0.19 ***	0.23 ***	0.23 ***	0.17 ***	0.18 ***	−0.09 ***	−0.09 ***	−0.05 ***	−0.05 ***
Employment status (ref. non-employed)												
Employed	0.73 ***	0.73 ***	0.53 ***	0.54 ***	0.95 ***	0.95 ***	1.00 ***	1.00 ***	−0.35 ***	−0.35 ***	−0.32 ***	−0.32 ***
Other	0.73 ***	0.73 ***	0.66 ***	0.66 ***	0.94 ***	0.94 ***	1.15 ***	1.15 ***	−0.26 ***	−0.26 ***	−0.31 ***	−0.31 ***
Marital status (ref. single)												
Married/cohabiting	0.13 ***	0.14 ***	0.19 ***	0.19 ***	0.72 ***	0.72 ***	1.07 ***	1.07 ***	−0.18 ***	−0.18 ***	−0.21 ***	−0.21 ***
Non-cohabiting partner	0.16 ***	0.16 ***	0.22 ***	0.22 ***	0.53 ***	0.53 ***	0.87 ***	0.87 ***	−0.07 ***	−0.07 ***	−0.13 ***	−0.13 ***
Constant	1.08	1.08	1.29	1.29	5.93	5.92	5.90	5.90	2.37	2.37	2.09	2.09
R^2^	0.1460	0.1461	0.0949	0.0950	0.1114	0.1115	0.1242	0.1243	0.1610	0.1611	0.1337	0.1338

*** *p* < 0.001, ** *p* < 0.01, * *p* < 0.05.

## Data Availability

The data underlying the findings is not freely available because of ethical and legal restrictions. However, data can be obtained via application at www.helsedata.no.

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
