# Peer review of "Caregiving + Migrant Background = Double Jeopardy? Associations between Caregiving and Physical and Psychological Health According to Migrant Backgrounds in Norway"

_ijerph, 2023, doi:10.3390/ijerph20105800_

Round 1

Reviewer 1 Report

It is an exciting study on the possible differences in health (physical and psychological) between native and migrant caregivers. This type of study is essential because it helps to visualize the situation in which the migrant population of a country lives concerning such a crucial issue as health. Below I make some suggestions for the text.

1. Although the study cannot precisely establish a causal relationship because of the limited data available, I believe the authors make a good analysis. In that sense, perhaps it would be better to present a study that does not analyze the relationship between being a caregiver and being a migrant but instead analyses the relationship between being a caregiver and having more or less health without correlating the migration variable. This recommendation is based on the limitations the authors mention in the text. For example, the questionnaire is in Norwegian, and we do not know if the migrants speak Norwegian, which limits their access and responses. In conclusion, establishing relationships between being a caregiver and having better or worse mental and psychological health would already be a good result. 

2. If the study goes on to compare migrants/natives, I suggest adding a paragraph with data on some differences between the native and migrant populations in Norway in the Introduction. For example, are there differences in health, education, employment, wages, etc.? This would help to justify the suspicion that there is also a difference in caregivers' mental and physical health concerning being a migrant or not.

3. You could also compare the two counties or age groups or other indicators such as rural and urban areas.

4. The sample comprises 109,934 (84.7%) natives versus 19,928 (15.3%) migrants. Could this disproportionality affect the results?

5. When the authors say "existing empirical literature" in line 72, are they referring to studies 25, 26, and 27 that are cited later in the same paragraph? If they are other studies, they should be cited.

6. The authors point out that one of the study's limitations is that the data were collected during the pandemic. In February 2021, there could be a more significant impact. However, in the second survey, in November 2021, most of the population was vaccinated, and there was a relative return to normality.

7. Regarding the Western and non-Western divisions, I have some doubts: 

- You consider Oceania as Western. So, does it include only New Zealand and Australia, or does it include the other countries (Fiji, Kiribati, Mashall, Solomon Islands, Nauru, Palau, Papua New Guinea, Tonga, Samoa, etc.)?

- On the other hand, is Latin America considered non-Western?

8. I would like to know if the study works with the variable gender or sex. Gender (as a social construct) is now understood from a much broader perspective than the traditional binary and dichotomous categorization of male or female, which excludes some transgender people. In contrast, the dichotomous approach presented in the paper is better explained through the variable sex (assigned at birth). The authors note in line 184 that "Between the two sexes there are similarities and differences," while at other points, they mention the gender variable. I suggest correcting that. On the other hand, if they work with "gender," it would be worth clarifying: 1) whether in the questionnaire people could self-determine their gender, 2) whether they had the space to self-determine other genders, and 3) whether anyone chose a gender other than male and female.

9. The authors note in lines 251-252 that "The findings contrast with those of previous literature supporting a double jeopardy hypothesis." I recommend mentioning one of those studies.

10. The authors note in lines 252-253 that "Various aspects of the Norwegian welfare system (e.g., support for work-family balance, formal care), shown to predict less caregiver distress in Norway than elsewhere." Considering that this is a comparative study with a migrant population, I would like to know if there is any data on the access of the migrant population to the "Norwegian welfare system." In other words: is it proven that migrants benefit as much as natives from this system, or what are the differences?

Author Response

  1. Although the study cannot precisely establish a causal relationship because of the limited data available, I believe the authors make a good analysis. In that sense, perhaps it would be better to present a study that does not analyze the relationship between being a caregiver and being a migrant but instead analyses the relationship between being a caregiver and having more or less health without correlating the migration variable. This recommendation is based on the limitations the authors mention in the text. For example, the questionnaire is in Norwegian, and we do not know if the migrants speak Norwegian, which limits their access and responses. In conclusion, establishing relationships between being a caregiver and having better or worse mental and psychological health would already be a good result.

Reply: Thanks for raising this interesting point. We agree that examining associations between caregiving and various health and well-being outcomes is important. Especially, we might add, as governments in many Western countries aim to stimulate and support increased informal care in response to population aging. These developments highlight the importance of exploring the potential costs of these aims for family members proving such care). A substantial literature already exists on these overall associations, and, in fact, we are currently working on a separate paper that examines longitudinal and multidimensional effects of caregiving overall without the migrant perspective.   

However, additionally exploring the associations with a focus on differences between migrants and non-migrants is important for several reasons, e.g. due to the growing migrant population across Western societies and its possible idiosyncratic needs, contexts, and consequences relating to caregiving (as explained in the paper). Also, while several stakeholders (e.g., carer alliances) and scholars have raised concerns and called for empirical studies on migrant carers, this research is still in its infancy and more knowledge is needed. While we agree that there are important caveats and limitations, as in most research, these should be addressed (as we have done) rather than discourage investigation. Especially in view of the lack of research in this field adopting a migrant perspective, we believe this novel study, based on uniquely large population-based data, represent an important first step towards recognizing the experiences of caregivers with a migrant background in a Western country.

  1. If the study goes on to compare migrants/natives, I suggest adding a paragraph with data on some differences between the native and migrant populations in Norway in the Introduction. For example, are there differences in health, education, employment, wages, etc.? This would help to justify the suspicion that there is also a difference in caregivers' mental and physical health concerning being a migrant or not.

Reply: This point is well taken. These points were addressed briefly in the introduction (lines 86-89), but we have now expanded this section with more evidence specifically from Norway regarding inequalities in socioeconomic resources, employment, and physical and mental health (lines 89-93).

  1. You could also compare the two counties or age groups or other indicators such as rural and urban areas.

Reply: There are limitations in the data as to which factors can be examined in moderator analysis (beyond the independent variables of the study), but we have in ancillary sensitivity analyzes examined the robustness/stability of the findings across counties and age groups, and the results are highly similar. This is now commented briefly under Results.

  1. The sample comprises 109,934 (84.7%) natives versus 19,928 (15.3%) migrants. Could this disproportionality affect the results?

Reply: We do not believe so. Both samples are large and, crucially, the migrant sample is very large compared to those in the reviewed extant literature. What is more important is the representativeness, and we have discussed potential biases in the end of the Discussion.

  1. When the authors say "existing empirical literature" in line 72, are they referring to studies 25, 26, and 27 that are cited later in the same paragraph? If they are other studies, they should be cited.

Reply: True, we refer to studies 25-27 in line 72. We agree regarding the lack of clarity here. To make in clearer we start sentence to with “For example….” (and later, “Similarly..”) to indicate that line 72 builds on the evidence that follows.  

  1. The authors point out that one of the study's limitations is that the data were collected during the pandemic. In February 2021, there could be a more significant impact. However, in the second survey, in November 2021, most of the population was vaccinated, and there was a relative return to normality.

Reply: This is a good point that deserves some mentioning in the paper. We now have included this potential caveat under Limitations, and we also refer to the aforementioned notion that findings are similar across countries, thus suggesting that timing vis-à-vis the pandemic is not highly impactful. Also, as shown in a government report (2022) on the timeline and developments about COVID-19 in Norway, Feb 2021 and Nov 2021 were quite similar in that both periods were considered peak periods of the pandemic (albeit with higher vaccination rates in the latter).

The Norwegian Government. (2022). Timeline: News from Norwegian Ministries about the Coronavirus disease Covid-19. Retrieved June 10 2022 from https://www.regjeringen.no/en/topics/koronavirus-covid-19/timeline-for-news-from-norwegian-ministries-about-the-coronavirus-disease-covid-19/id2692402/

  1. Regarding the Western and non-Western divisions, I have some doubts:

- You consider Oceania as Western. So, does it include only New Zealand and Australia, or does it include the other countries (Fiji, Kiribati, Mashall, Solomon Islands, Nauru, Palau, Papua New Guinea, Tonga, Samoa, etc.)?

- On the other hand, is Latin America considered non-Western?

Reply: Data that were used in this study had predefined categories of regions, that we again recoded into western and non-western migrant groups. It is correct that Oceania is considered as western in this study, simply because this category was merged with North America in the data set (“Oceania/North America”). Regarding the other countries listed above, there are few if any individual from these countries living in Norway and the probability of them participating in the surveys is minimal. “Latin America and Caribbean” is also a predefined category, and insofar as we have individuals from this region they are categorized as “non-Western”. This is in accordance with the definition of “Western” and “Non-Western” (all except Western Europe, North America, and Oceania) that we use and that have routinely been used by Statistics Norway, and others. We now describe this categorization in more detail.

  1. I would like to know if the study works with the variable gender or sex. Gender (as a social construct) is now understood from a much broader perspective than the traditional binary and dichotomous categorization of male or female, which excludes some transgender people. In contrast, the dichotomous approach presented in the paper is better explained through the variable sex (assigned at birth). The authors note in line 184 that "Between the two sexesthere are similarities and differences," while at other points, they mention the gender variable. I suggest correcting that. On the other hand, if they work with "gender," it would be worth clarifying: 1) whether in the questionnaire people could self-determine their gender, 2) whether they had the space to self-determine other genders, and 3) whether anyone chose a gender other than male and female.

Reply: Again, a point well taken. Information about sex was extracted from the national register, where legal sex is reported and that consists of two categories: male and female. Thus, respondents could not self-determine their gender when filling out the questionnaire. We now have changed from “gender” to “sex” consistently in the paper.

  1. The authors note in lines 251-252 that "The findings contrast with those of previous literature supporting a double jeopardy hypothesis." I recommend mentioning one of those studies.

Reply: Agree, we now have included the relevant references here.

  1. The authors note in lines 252-253 that "Various aspects of the Norwegian welfare system (e.g., support for work-family balance, formal care), shown to predict less caregiver distress in Norway than elsewhere." Considering that this is a comparative study with a migrant population, I would like to know if there is any data on the access of the migrant population to the "Norwegian welfare system." In other words: is it proven that migrants benefit as much as natives from this system, or what are the differences?

Reply: This is an important point that unfortunately is mostly left to speculation due to lack of data. We have inserted a sentence (after lines 252-253) which underscores that the hypothesis (that welfare supports may alleviate caregivers distress among migrants) remains speculation before we have data on the use and benefits of the formal services and supports for migrant caregivers. We also add some mentioning and discussion of the fact that migrants use formal care services to a lower extent than non-migrants in Norway.

Reviewer 2 Report

Interesting theme approached, well backgrounded and written.

Suggestions: 

- Reduce keywords to 5

- in line 148/149 clarify the reason for adopting this age categorization;

- in lines 180-182 is mentioned"there’s a gradually rising gradient in mental health and psychological well-being from non-migrants to Western migrants to non-Western migrants, although most differences are not statistically significant". Is important to highlight which are statistically significant.

- Both in results and discussion, objective results and significance levels are needed to understand the study results.

- table 2 doesn't identify p values.

Author Response

  1. Reduce keywords to 5

Reply: Done.

  1. In line 148/149 clarify the reason for adopting this age categorization;

Reply: We now clarify that we merely adopted the age categorization that existed in the data we were provided. The data provider provided age in intervals and not in years to avoid identification of respondents based on information about age in combination with other data.

  1. In lines 180-182 is mentioned "there’s a gradually rising gradient in mental health and psychological well-being from non-migrants to Western migrants to non-Western migrants, although most differences are not statistically significant". Is important to highlight which are statistically significant.

Reply: Agree. This is now clarified.

  1. Both in results and discussion, objective results and significance levels are needed to understand the study results.

Reply: We have now reviewed these sections to clarify and indicate results and significance levels.

  1. Table 2 doesn't identify p values.

Reply: Thanks for pointing out this omission. This is now inserted.

Round 2

Reviewer 1 Report

Thank you for your attention to the suggestions. The important thing is that you thought about these elements before the article was published. That would be all for my part.

Best regards